# The Mitochondrial Biomarkers FGF-21 and GDF-15 in Patients with Episodic and Chronic Migraine

**DOI:** 10.3390/cells10092471

**Published:** 2021-09-18

**Authors:** Philipp Burow, Marc Haselier, Steffen Naegel, Leila Motlagh Scholle, Charly Gaul, Torsten Kraya

**Affiliations:** 1Department of Neurology, University Hospital Halle-Saale, Ernst-Grube-Straße 40, 06120 Halle (Saale), Germany; marc-haselier@web.de (M.H.); steffen.naegel@medizin.uni-halle.de (S.N.); leila.scholle@uk-halle.de (L.M.S.); Torsten.Kraya@sanktgeorg.de (T.K.); 2Headache Center Frankfurt, Dalbergstraße 2A, 65929 Frankfurt am Main, Germany; c.gaul@kopfschmerz-frankfurt.de; 3Department of Neurology, Hospital Sankt Georg, Delitzscher Straße 141, 04129 Leipzig, Germany

**Keywords:** mitochondrial disorder, headache, migraine, FGF-21, GDF-15

## Abstract

Mitochondrial processes may play a role in the pathophysiology of migraine. Serum levels of two biomarkers, Fibroblast-growth-factor 21 (FGF-21) and Growth-differentiation-factor 15 (GDF-15), are typically elevated in patients with mitochondrial disorders. The study investigated whether the presence of migraine may influence FGF-21 and GDF-15 serum levels considering vascular and metabolic disorders as possible confounders. A cross-sectional study in two headache centers was conducted analyzing GDF-15 and FGF-21 serum concentration in 230 patients with episodic and chronic migraine compared to a control group. Key clinical features of headache were evaluated, as well as health-related life quality, anxiety and depression using SF-12 and HADS-questionnaires. Elevated GDF-15 values were detected in the migraine group compared to the control group (506.65 ± 275.87 pg/mL vs. 403.34 ± 173.29 pg/mL, *p* < 0.001, Mann–Whitney U test). A strong correlation between increasing age and higher GDF-15 levels was identified (*p* < 0.001, 95%-CI elevation of GDF-15 per year 5.246–10.850 pg/mL, multiple linear regression). Mean age was different between the groups, and this represents a confounding factor of the measurements. FGF-21 levels did not differ between migraine patients and controls (*p* = 0.635, Mann–Whitney U test) but were significantly influenced by increasing BMI (*p* = 0.030, multiple linear regression). Neither biomarker showed correlation with headache frequency. Higher FGF-21 levels were associated with a higher mean intensity of headache attacks, reduced health-related life quality and anxiety. When confounding factors were considered, increased serum levels of FGF-21 and GDF-15 were not detected in migraine patients. However, the results show an age-dependence of GDF-15 in migraine patients, and this should be considered in future studies. Similar findings apply to the relationship between FGF-21 and BMI. Previous studies that did not adjust for these factors should be interpreted with caution.

## 1. Introduction

Clinical, biochemical and genetic studies indicate the role of mitochondrial dysfunction in the pathogenesis of migraine. It is a complex disorder of sensory processing, and its pathophysiology is far from being fully understood [1]. As migraine prevalence is approximately three times higher in women than in man, a possible mitochondrial involvement in the pathophysiology of migraine has been discussed for many years. 

In this context, biochemical analyses in patients with migraine showed significantly elevated lactate levels in serum and liquor pointing towards a disturbance in the oxidative metabolism [2,3]. This hypothesis was corroborated by decreased activity levels of respiratory chain enzymes in the serum of patients with migraine [4]. Using invasive diagnostic methods, ragged-red fibers and COX-negative (Cytochrom-c-oxidase negative) fibers as typical histological changes in mitochondrial diseases were found in muscular biopsies of patients with migraine with aura [4]. In MR-spectroscopy, ATP was found to be significantly decreased in the occipital cortex of patients with migraine with aura [5]. Phosphate metabolism was altered in migraine patients without aura during an attack, as well as in the occipital lobes [6]. Controlled studies showed the efficacy of riboflavin (vitamin B2) and coenzyme Q10 as modulators of the mitochondrial respiratory chain in migraine prophylaxis [7,8]. A possible pathophysiological connection between MELAS (mitochondrial encephalopathy, lactic acidosis and stroke-like episodes) and migraine has recently been proposed [9]. Disturbances of oxidative phosphorylation may lead to increased oxidative stress and a disturbed cellular redox state. Vascular endothelial and smooth muscular cells respond with increased mitochondrial biogenesis, eventually leading to narrowing of the vessel lumen. Increased shear and wall stress may trigger platelet aggregation. The consequent hypoxia/ischemia, altered glutamate metabolism, and ionic homeostasis may trigger cortical spreading depolarization (CSD), which may then activate the trigemino-vascular system leading to headache [10]. However, it is unclear which signaling molecules are exactly involved in these processes. These findings, among others, indicate an involvement of oxidative metabolism in migraine during and possibly between attacks. Thus far, however, the question cannot be answered whether mitochondrial dysfunction is involved in the development of migraine attacks, whether recurrent migraine attacks induce mitochondrial dysfunction, or whether both are present. 

From the epidemiologic point of view, several studies show a higher prevalence of migraine than expected in patients with a genetically proved mitochondrial disorder such as MELAS, MERRF (myoclonic epilepsy with ragged red fibers) and LHON (Leber’s hereditary optic neuropathy) [11,12,13,14]. Migraine is a substantial contributor for burden of disease in patients with mitochondrial diseases [15]. However, in genetic analyses, none of the frequent point mutations of the mitochondrial DNA, e.g., m.3243 A > G associated with the MELAS-syndrome, in patients with migraine with aura were detected [16]. Zaki et al. found an association between polymorphisms of the mitochondrial DNA and migraine (m.16519C > T and m.3010G > A) [17]. A recent mitochondrial genome-wide study did not show associations of any mitochondrial variant or haplotype with migraine and thus could not replicate the findings from some of the previous studies [18]. The different results from epidemiologic and genetic studies lead to the question of whether there are other suitable methods to investigate a mitochondrial dysfunction in patients with migraine.

As non-invasive and economical steps in the diagnosis of a mitochondrial disease, the peptide molecule fibroblast-growth-factor-21 (FGF-21) and growth-differentiation-factor-15 (GDF-15) are established biomarkers. Both were found to be elevated in patients with mitochondrial diseases [19,20,21]. Their sensitivity and specificity for mitochondrial diseases were found to be higher than other metabolic parameters such as lactate, pyruvate, the lactate-pyruvate ratio or creatine kinase. FGF-21 seems to play a significant role in energy homeostasis regulating the metabolism of glucose and fatty acids [22,23]. It is also discussed as having anti-inflammatory and neuroprotective capacities [24]. Elevated levels are found especially in patients with mitochondrial myopathy [25,26]. In contrast, the physiology of GDF-15 is far less understood. With regard to the diagnosis of mitochondrial diseases, GDF-15 seems to have an improved sensitivity and specificity in comparison to FGF-21 [27,28]. It is a member of the transforming growth factor β-superfamily and can be induced in a p53-dependent manner by cellular stress [24], e.g., because of ischemia. Elevated GDF-15 levels are linked to chronic inflammation, oxidative stress and tissue damage [29]. It may have cardioprotective effects [30]. An anti-inflammatory role in inflammatory diseases, e.g., rheumatoid arthritis or atherosclerosis, is also discussed [31]. 

This study investigated whether serum concentrations of both FGF-21 and GDF-15 are altered in patients suffering migraine considering disease severity, attack morphology and concomitant disorders.

## 2. Materials and Methods

### 2.1. Study Description and Recruitment

A multi-center controlled cross-sectional study was conducted. All participants were recruited at two tertiary headache centers for all participants suffering migraine and at an outpatient medical office for control subjects. Subjects were eligible if aged between 18 and 70 years. For the migraine group, patients with episodic migraine (EM) with at least one migraine day a month within the last three months and chronic migraine (CM) were eligible. A well-kept headache diary at least for the last three months was necessary for inclusion. The presence of other primary or secondary headache disorders other than tension-type headache (maximum three days per month) was an exclusion criterion in this group. In the control group, an infrequent tension-type headache with no more than one headache day a month was allowed. Headache diagnoses were made by a headache expert at each center. A physical and clinical neurological examination was performed as part of the clinical routine. 

General exclusion criteria were (a) no informed consent could be taken and (b) at least one of the following criteria is met: proved or supposed mitochondrial disorder, obesity (BMI ≥ 30 kg/m^2^), myopathy, diabetes mellitus, known coronary heart disease and liver disease. If there were anamnestic (e.g., cardiomyopathy, visual disturbances at a young age, epilepsy of unknown origin) or clinical–neurological (e.g., oculomotor disturbance of unclear origin) indications of a possible mitochondrial disease, the patient was not included in the study. The local ethics committee of the Martin-Luther-University Halle-Wittenberg approved the study. Written informed consent was obtained from all individual participants included in the study. Data collection took place in the years 2018 and 2019.

### 2.2. Headache Characteristics and Psychometric Questionnaires

At the headache departments, a structured headache interview was conducted with a standardized headache questionnaire developed for this study. It based on the criteria for migraine of the beta version of the International classification of headache disorders (ICHD-3ß) [32]. Headache and migraine frequency (days a month), headache quality, localisation, average and maximum intensity, concomitant symptoms, the behaviour during a headache attack, trigger factors, the presence of aura symptoms and the family history of headache were addressed in particular. Headache intensity was classified into groups with mild (1–3), moderate (4–6) and severe (7–10) intensity based on the numeric rating scale (NRS). Furthermore, frequency and efficacy of current and past medication for acute and preventive treatment of migraine were collected. A headache-specialized neurologist conducted all interviews. Furthermore, participants in the migraine group were asked to fill out MIDAS (Migraine Disability Assessment), SF-12 (Short-Form 12) and HADS-D (Hospital Anxiety and Depression Scale) questionnaires [33,34,35]. For interpretation of HADS-D, a cut-off value of 8 was used in both subscales to distinguish patients with normal (0–7 pts.) from at least conspicuous symptoms. This group composed of conspicuous (8–11 pts.), severe (12–14 pts.) and very severe (15–21 pts.) symptomatology according to [33].

### 2.3. Plasma Concentrations of FGF-21 and GDF-15

Plasma concentrations were calculated using an ELISA-based approach. After the headache interview, blood samples were drawn from a peripheral vein (Cubital vein if possible) into a 9 mL tube prefilled with EDTA with a final concentration of 1.6 mg/mL (Sarstedt AG, Nümbrecht, Germany). Blood samples were centrifuged at 2500× *g* for 15 min and plasma was then transferred to polypropylene tubes and stored at −80 °C. Concentrations of FGF-21 and GDF-15 were measured using commercially available kits (Bio Vendor Human FGF-21 ELISA, Bio Vendor Human GDF-15 ELISA; Bio Vendor, Brno, Czech Republic). The scientist who performed the assays (L.S.) was blinded to the study group the sample belonged to.

### 2.4. Statistical Analysis

Explorative statistical analyses were performed using IBM SPSS Statistics 25.0 for Windows (IBM Corp., New York, NY, USA). Correlations between the concentrations of the mitochondrial biomarkers and parameters of headache were analyzed using simple and multiple linear regression. For non-dichotomous characteristics such as age, grouped analyses were performed as shown in the respective sections. Non-parametric tests (Mann-Whitney U test and Kruskal-Wallis test) were used to compare FGF-21 and GFD-15 between groups. The individual statistical tests are indicated within the results section. 

## 3. Results

### 3.1. Composition and Description of Study Groups 

A total of 328 patients were included in the study, 230 in the migraine group and 98 in the control group. Demographic data and headache-related parameters are presented in Table 1. Within the migraine group, in 47.8%, a chronic migraine (CM) was diagnosed according to ICHD-3ß. The mean headache burden was 15 days per month in the whole migraine group. Data regarding migraine prophylaxis were evaluable for 68% (156/230) of the migraine patients. For acute medication, data were available from 54% (125/230) of migraine patients. Medication-overuse headache (MOH) could be diagnosed in 14% of patients based on ICHD-3ß criteria.

The MIDAS score showed a severe disability in 207 patients (91.6%). The results of the SF-12 questionnaire were compared with a norm sample representative for Germany [34]. The physical sum scale in the migraine group was 38.65 ± 9.06, and the mental sum scale 42.39 ± 10.92. Thus, 85% of the study group had more severe physical impairment and 80% more severe mental impairment than the German norm sample. In the depression section of the HADS-D questionnaire, 21.0% of EM patients had considerable symptoms; in the CM group, 44.6% did. In the anxiety section, the corresponding proportions were 42.8% and 53.6%, respectively. 

### 3.2. Influence of Migraine on the Biomarkers

All data are given as median and 25–75% quartile range unless otherwise specified. The plasma level of FGF-21 was not different comparing the migraine group (27.5 pg/mL, 16.3–85.5 pg/mL, *n* = 230) with the control group (38.0 pg/mL, 14.9–94.0 pg/mL, *n* = 98, *p* = 0.635, Mann–Whitney U test). In contrast, GDF-15 plasma levels were elevated in the migraine group (468.4 pg/mL, 348.3–606.1 pg/mL, *n* = 226) compared to controls (362.5 pg/mL, 287.0–493.3 pg/mL, *n* = 96, *p* < 0.001, Mann–Whitney U test). No differences were seen comparing EM with CM with regard to both FGF-21 (*p* = 0.195) and GDF-15 (*p* = 0.924, Mann–Whitney U test). The results are summarized in Figure 1. 

Furthermore, no correlation could be observed between the concentration of any biomarker and the number of headache days a month, the localization of headache, the presence of aura and family history of migraine (Table 2). Regarding the average headache intensity, a grouped analysis with three pain levels (NRS 1–4, 5–7 and 8–10) was performed. It shows increased FGF-21 levels in patients with a higher pain intensity (*p* = 0.042, Kruskal–Wallis test). This could not be found for GDF-15 (*p* = 0.398, Kruskal–Wallis test). The results with the appropriate statistical tests are summarized in Table 2. The GDF-15 values of four participants of the control group and two patients from the migraine group were excluded from the statistical analysis as they could not be determined precisely due to technical problems.

### 3.3. Association of Age and GDF-15 Levels 

For GDF-15, a strong association of higher values with increasing age was found. Participants were divided into age cohorts (10 years). A difference of GDF-15 values between all groups could be shown (Kruskal-Wallis test, *p* < 0.001, Figure 2a). This was the case for all participants, but also for the migraine and the control group separately (Kruskal–Wallis tests, *p* < 0.001). In the multiple regression analysis considering possible influencing factors (Table 3), a strong correlation between increasing GDF-15 values and increasing age was shown (*p* < 0.001, 95% confidence interval: 6.62–10.82 (pg/mL)/year). This was reproducible for both recruitment sites. Comparing GDF-15 levels between the control and the migraine group within the defined age groups, no significant differences could be detected (*p*-range 0.100–0.512, Mann-Whitney U tests).

### 3.4. Association of BMI and FGF-21 Levels 

FGF-21 levels raise with increasing BMI. For analysis, the participants were divided in BMI groups (Figure 2b). The statistical analysis shows a difference of FGF-21 values between the groups (*p* < 0.001, Kruskal-Wallis test). This was true for both recruitment sites. In the multiple regression analysis (Table 3), a positive correlation between FGF-21 levels and increasing BMI can be shown (*p* = 0.017, 95% confidence interval: 0.79–7.81 (pg/mL)/BMI). This was reproducible for both recruitment sites. Within the defined BMI groups as stated in Figure 2b, no significant differences in FGF-21 levels could be detected between control and migraine group (*p*-range 0.188–0.719, Mann-Whitney U tests).

### 3.5. Multiple Regression Analysis 

Multiple regression analysis was performed including potentially influencing factors on the FGF-21 and GDF-15 values collected in this study. All parameters included in the regression analysis are shown in Table 3. For a better overview of the results, only *p*-values with the corresponding confidence intervals and not the medians are shown. The analysis confirmed the association between BMI and FGF-21 and between age and GDF-15 when the other factors are also included. In addition, the results showed that higher headache intensity was associated with higher levels of both biomarkers. However, taking all these factors into account, no association between the presence of migraine and elevated serum concentrations of the two mitochondrial biomarkers can be demonstrated, as shown in the first row in Table 3. Regarding quality of life and anxiety, the presence of anxiety symptoms according to HADS-D was associated with lower FGF-21 levels. This was not true for the depression part. Furthermore, improved physical health according to SF-12 was associated with lower FGF-21 levels. For GDF-15, no association between the MIDAS questionnaire, any parts of HADS-D and SF-12 could be detected. These results, calculated as multiple regression, are shown in Table 4. 

## 4. Discussion

This study investigated the association between serum concentrations of the biomarkers FGF-21 and GDF-15, which have been investigated for mitochondrial diseases, and the presence of migraine. In a non-parametric analysis, elevated serum levels of GDF-15, but not FGF-21, were found in a large cohort of patients with migraine compared with an appropriately sized control group. The migraine group consisted of nearly 50% of patients with chronic migraine and therefore represents a group of severely affected patients at two tertiary headache centers. This is underlined by the results of SF-12 and HADS-questionnaire in the migraine group showing reduced life quality and depressive symptoms in a relevant proportion of the patients. 

However, in the multiple regression correcting for identified confounders, no association between both biomarkers and migraine was found. They are peptide molecules with pleiotropic effects. Although both parameters (possibly GDF-15 superior to FGF-21) are considered capable of distinguishing patients with mitochondrial diseases from those without [36], serum concentrations of both molecules are likely to be significantly affected by vascular and metabolic diseases. GDF-15 is considered a stress-induced cytokine, and it may play a role in various pathophysiological processes, e.g., heart insufficiency, myocardial infarction, pulmonary hypertension, diabetes, metabolic syndrome, autoimmune diseases and cancer [37,38]. An association of higher GDF-15 levels and increasing age has also been described [39,40,41]. This is confirmed by our data. As patients with severe vascular and metabolic diseases were excluded from the study (see Methods), increased GDF-15 levels are probably due to the different age between the two groups (mean difference 7 years). In our analysis, this seems to play a role especially for patients 40 years and older. Regarding FGF-21, serum levels are typically elevated in patients with obesity [42] and thus it was an exclusion criterion. Nonetheless, a significant association between BMI and FGF-21 was observed even in our study group with patients of normal weight or slightly overweight. FGF-21 levels are also affected by fasting and physical activity [43].

Previous studies have identified different cut-off values of the two biomarkers for mitochondrial disease in children, young adults, and adults [14,15,16]. Even considering the different kits for the determination of the biomarkers, the values especially for GDF-15 in this study were markedly below the cut-off values. 

Regarding clinical characteristics of migraine, no association between the two biomarkers and monthly headache days and no difference between patients with episodic and chronic migraine were detected. In this context, it has already been shown that patients with mitochondrial diseases and migraine do not have an increased attack frequency compared to patients with migraine [12]. With increasing average headache intensity, serum concentrations of both FGF-21 and GDF-15 were slightly higher. The clinical significance remains unclear at present. In this study, a present headache attack during the determination of the biomarkers was not considered. A longitudinal approach may help to understand if a recent and strong migraine attack itself leads to the secretion of one or both biomarkers. 

No associations with depression, anxiety or decreased quality of life were found for GDF-15 in this study. Anxiety correlates with lower FGF-21 values in migraine patients. This is partially in line with existing data showing reduced FGF-21 levels in cerebrospinal fluid in patients with anxiety symptoms [44], although a corresponding connection with depressive symptoms has also been described here. However, a larger study with a more detailed assessment of depression and anxiety symptomatology (the questionnaires used here serve more as a screening method) would be necessary to validate these relationships.

Our data underline that
(a)The determination of FGF-21 und GDF-15 does not allow a valid statement about mitochondrial involvement in migraine. This may also be true in other diseases where mitochondrial involvement is suspected. This is in line with a recently published study, in which plasma levels of GDF-15 assessed in patients with open angle glaucoma were found to be strongly confounded by age and vascular diseases [41]. Due to the extremely diverse factors influencing serum concentrations of both FGF-21 and GDF-15, it is difficult to define an independent control group.(b)Studies that investigated FGF-21 and GDF-15 and did not correct for vascular, inflammatory and metabolic disease should be interpreted with caution.(c)Mitochondrial involvement in migraine cannot be excluded. There are still no known biomarkers in migraine that can predict, e.g., a therapeutic response to prophylaxis and/or acute therapy. Further studies, preferably in a younger group, could investigate the effect of vitamin B2 (riboflavin) and coenzyme Q 10 in migraine patients with elevated GDF-15 and/or FGF-21 levels.

Limitations of this work are the exclusive cross-sectional approach and the use of a non-age-matched control group. The biomarkers were collected at different times of the day, and the subjects did not have to be fasting. Life circumstances, e.g., food habits or physical activity, were not considered. Furthermore, metabolic or cardiac diseases were not recorded in detail (e.g., arterial hypertension according to WHO grades), except for the exclusion criteria mentioned above. Chronic inflammatory diseases may affect GDF-15 levels, as well as possibly FGF-21, and this was not analyzed in detail. Additionally, despite history and clinical examination, mono- or oligosymptomatic mitochondrial diseases in particular cannot be excluded in the study population. The influence of long-term medication, e.g., migraine prophylaxis, on biomarker values was not studied explicitly. This could be further investigated in a longitudinal study. From the group of anticonvulsants, valproate in particular should be avoided in mitochondrial diseases [45,46]. Since this affects only a very small part of the study population, the possible influence on the biomarker levels, if any, should be small. In addition, there are other biochemical and genetic options to investigate mitochondrial involvement, which were not carried out in this study. It is currently not fully understood whether FGF-21 and GDF-15 are altered in mitochondrial diseases predominantly with specific organ involvement (e.g., mitochondrial myopathies) or more generally indicate mitochondrial dysfunction. 

The strengths of this work are the well-characterized migraine group with detailed recording of clinical parameters at two tertiary headache centers, the analysis of anxiety and depression symptomatology and the recording of health-related quality of life.

A further diagnostic of mitochondrial diseases will be offered to the participants with highly elevated levels of one or both biomarkers.

## Figures and Tables

**Figure 1 cells-10-02471-f001:**
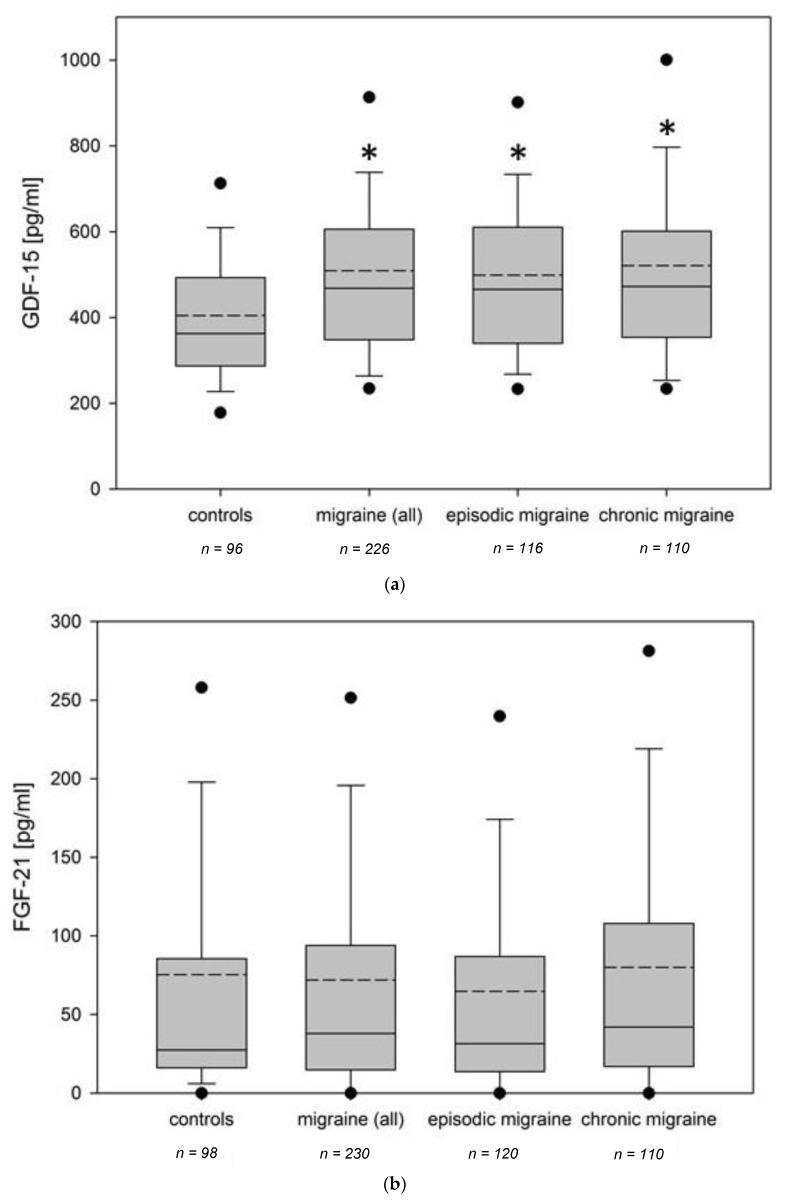
FGF-21- and GDF-15 values within the study groups. (**a**) FGF-21 and (**b**) GDF-15 values of all migraine patients, subgroups (EM and CM) and control subjects. Within the box plots, solid lines represent medians and dashed lines means. Outliers are displayed as 5–95% range (black dots). Groups that differed significantly (*p* < 0.05) from the control group are marked with an asterisk.

**Figure 2 cells-10-02471-f002:**
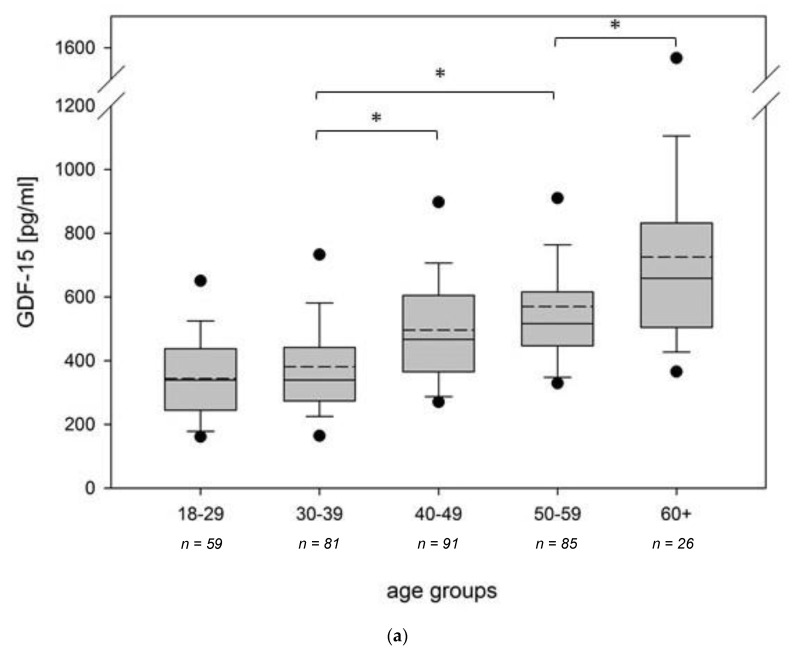
FGF-21- and GDF-15 values depending on age or BMI. (**a**) GDF-15 serum concentration within different age groups, (**b**) FGF-21 serum concentration within different BMI groups. Within the box plots, solid lines represent medians and dashed lines means. Outliers are displayed as 5–95% range (black dots). Significant differences (*p* < 0.05) between the groups using pairwise multiple comparison on ranks (Dunn’s method) are marked with an asterisk.

**Table 1 cells-10-02471-t001:** Demographic and headache-related parameters of the study groups. Unless otherwise stated, values are presented as mean ± standard deviation (SD). Regarding demographic parameters, migraine and control groups were compared using the tests indicated in the table. Monthly headache days were averaged using headache diaries in the last three months before study inclusion. This was also carried out for the attack intensities. When migraine with aura was diagnosed, attacks without aura could also occur in the same patient. For migraine prophylaxis, only data on medication-based regimens were evaluated. CM: Chronic migraine; EM: Episodic migraine; MOH: Medication-overuse headache.

Characteristic	Migraine Group (*n* = 230)	Control Group (*n* = 98)	*p*-Value
age (years)	44.16 ± 9.10 range: 18–69	36.86 ± 10.60 range: 18–59	<0.001 (*t*-test)
gender (female/male)	206/24 (89% female)	71/27 (72% female)	0.001 (chi^2^-test)
BMI (kg/m^2^)	24.37 ± 3.57	23.96 ± 3.41	0.24 (*t*-test)
	**Migraine Group** **(*n* = 230)**		
migraine type (CM/EM)	110/120 (48% CM)		
migraine with/without aura	96/134 (42% with aura)		
average headache days (monthly)	whole group: 15.16 ± 9.21 (*n* = 230) CM: 22.79 ± 6.33 (*n* = 110) EM: 8.10 ± 4.74 (*n* = 120)		
average intensity of attacks (NRS 0–10)	6.09 ± 1.44		
family history for migraine	165 positive /75 negative (72% positive)		
medication-overuse headache (yes/no)	18/107 (14% MOH, *n* = 125)		
previous migraine prophylaxis (attempts)	1.42 ± 1.31 (*n* = 156)		
migraine prophylaxis present at the time of study participation (yes/no)	116/40 (74% yes, *n* = 156) 34% ß-blocker (metoprolol or bisoprolol) 32% amitriptyline or another antidepressant 17% anticonvulsant (14% topiramate, 2% valproate) 6% erenumab, 4% onabotulinumtoxin A, 1% flunarizine 6% other (e.g., candesartane)		

**Table 2 cells-10-02471-t002:** Clinical parameters of headache and their impact on GDF-15 and FGF-21 levels. The results are given as *p* values (*p* < 0.05 in bold print). At *p* values > 0.05, there is no statistically significant association between the respective clinical parameter and the biomarker. Due to different variable types, different statistical tests were used for analysis. Only for increasing headache intensity was a correlation with higher FGF-21 levels observed.

Clinical Parameters	GDF-15	FGF-21	Test Method Used
episodic vs. chronic migraine	0.924	0.195	Kruskal-Wallis test
headache days	0.837	0.267	simple linear regression
headache intensity (groups)	0.398	**0.042**	Kruskal-Wallis test
uni- vs. bilateral headache	0.303	0.506	Kruskal-Wallis test
positive vs. negative family history for migraine	0.103	0.459	Kruskal-Wallis test

**Table 3 cells-10-02471-t003:** Multiple linear regression of GDF-15 and FGF-21 concentrations *p*-values were calculated using multiple regression. Additionally, the 95% confidence intervals of the biomarker concentrations in pg/mL are given in brackets. For example, there is a highly significant correlation between age and GDF-15 with an increase in GDF-15 of 6.6–10.9 pg/mL per year. This type of reading applies to all metric-scaled parameters (headache days, age and BMI). Grouped analyses were performed for all other parameters. The *p*-values are stratified for each clinical parameter listed in this table.

Clinical Parameters	GDF-15 ^1^	FGF-21 ^1^
Migraine present yes/no	0.177 (−18.3–99.0)	0.564 (−32.2–17.6)
Episodic vs. chronic migraine	0.922 (−148.8–299.5)	0.107 (−6.3–64.1)
Headache days/month	0.925 (−5.9–6.5)	0.548 (−2.6–1.3)
Headache intensity (groups)	0.041 (1.0–48.8)	0.044 (0.2–15.2)
Uni- vs. bilateral headache	0.632 (−91.4–55.7)	0.236 (−9.3–37.3)
Family history for headache (pos. /neg.)	0.156 (−128.8–20.7)	0.983 (−23.5–24.0)
gender (female vs. male)	0.404 (−40.9–101.4)	0.160 (−52.1–1.9)
Age (years)	<0.001 (6.6–10.9)	0.120 (−0.2–1.6)
BMI	0.236 (−13.0–3.2)	0.017 (0.8–7.8)

**^1^** Values are given as *p*-value (95%-confidence interval [pg/mL]).

**Table 4 cells-10-02471-t004:** Multiple linear regression of GDF-15 and FGF-21 concentrations depending on the results of MIDAS, HADS and SF-12 questionnaires. For the calculations, all *p*-values are stratified for each clinical parameter as shown in Table 3. The results are presented in a similar way to Table 3. For example, higher values in the anxiety part of HADS-D significantly correlate with a decrease of FGF-21 between 3.6 and 36.0 pg/mL per point in HADS-D. All parameters were analyzed metrically. For interpretation, higher scores in the MIDAS and HADS sections are associated with higher impairment, while this is indicated by lower scores in the SF-12.

Questionnaire	GDF-15 ^1^	FGF-21 ^1^
MIDAS	0.909 (−70.6–62.8)	0.761 (−17.5–23.9)
HADS−D depression	0.316 (−29.0–89.3)	0.738 (−21.9–15.6)
HADS−D anxiety	0.158 (−88.4–14.5)	0.017 (−36.0−(−3.6))
SF−12 physical part	0.909 (−4.5–4.0)	0.019 (−2.9−(−0.263))
SF−12 mental part	0.626 (−5.2–3.1)	0.162 (−2.2–0.4)

**^1^** Values are given as *p*-value (95%-confidence interval [pg/mL]).

## Data Availability

The data presented in this study are available on reasonable request from the corresponding author.

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
