# Peer review of "The Mitochondrial Biomarkers FGF-21 and GDF-15 in Patients with Episodic and Chronic Migraine"

_cells, 2021, doi:10.3390/cells10092471_

Round 1

Reviewer 1 Report

The authors present an interesting study in which biomarkers commonly associated with mitochondrial disorders are investigated as potential markers for migraine, namely FGF-21 and GDF-15. In order to examine this, the authors utilised two medical centres and recruited those experiencing episodic and chronic migraines, surveying them as to the severity and frequency of their condition (amongst other factors), and also taking blood draws for serum analysis of the aforementioned biomarkers. Interestingly, the authors make a number of notable observation from their study, such as GDF-15 serum levels increases with age and FGF-21 levels correlate with BMI, however a number of confounding factors limit the power of data with respect to migraine presence/intensity.

In reviewing the manuscript I had a number of concerns. The following should be addressed by the authors in subsequent submissions:

  1. The language of the manuscript is good for the most part, however the flow could be improved by giving more detail to some points describing studies pertinent to the hypothesis. Some points are vague and non-descript and more details will address this. Also there are some small language/grammar errors within and the manuscript should be scanned to address these.
  2. Whilst a number of criteria were included for excluding certain data sets from the study, I was surprised by some which were not included such as whether the person was on any particular medications at the time of evaluation. In that way the study design is questionable and the authors should justify this design.
  3. When performing evaluation of the bloods, were all the participants recruited and bloods drawn and stored until all samples were gathered, or were bloods ran as soon as they were collected? If the former, how long were the bloods stored for before analysis was conducted?
  4. It would be useful if the n-number was included on each graph
  5. Were any statistical comparisons performed on the data presented? If so, some sort of annotation to indicate significant differences would be useful.
  6. Did the authors examine the difference between males and females at all?
  7. Are the surveys available online? Are they all validated for this kind of examination?

Author Response

- The language of the manuscript is good for the most part, however the flow could be improved by giving more detail to some points describing studies pertinent to the hypothesis. Some points are vague and non-descript and more details will address this. Also there are some small language/grammar errors within and the manuscript should be scanned to address these.

Thank you for your comment. We have revised the introduction in particular and checked the manuscript for grammatical errors.

- Whilst a number of criteria were included for excluding certain data sets from the study, I was surprised by some which were not included such as whether the person was on any particular medications at the time of evaluation. In that way the study design is questionable and the authors should justify this design.

This is an important issue. In this study, not all concomitant medications were recorded, but only the prophylaxis and acute therapy of migraine. Unfortunately, the data set was not complete in this regard, so we did not include the analyses in the manuscript. Discontinuation of migraine prophylaxis before the study, which would likely have improved the power, was not reasonable in our severely affected patient population. We consider an influence of acute therapy on biomarkers to be low because of the short half-life of the drugs (e.g., ibuprofen or sumatriptan).

We added the following data to the manuscript:

- The proportion of patients on migraine prophylaxis and the individual substance categories in the methods section                                                                              - A short discussion esp. on antiepileptic drugs/valproate (the latter concerns only a small proportion of patients) in the discussion section

When performing evaluation of the bloods, were all the participants recruited and bloods drawn and stored until all samples were gathered, or were bloods ran as soon as they were collected? If the former, how long were the bloods stored for before analysis was conducted?

After blood collection, the samples were immediately processed and frozen at -80°C as quickly as possible. Analysis by ELISA was performed maximum 3 months after collection, during phases with good recruitment mostly already within 4 weeks.

It would be useful if the n-number was included on each graph

Thank you for the comment. The graphs have been updated accordingly.

Were any statistical comparisons performed on the data presented? If so, some sort of annotation to indicate significant differences would be useful.

The results of pairwise comparisons between individual study groups were added to all graphs accordingly (Dunn's method).

Did the authors examine the difference between males and females at all?

A gender comparison for both biomarkers is included in Table 2 (as part of the multiple regression). The validity is very limited due to the small proportion of men in the study group. In a non-parametric comparison between males and females, the p-values are 0.744 for FGF-21 and 0.286 for GDF-15. Because of the higher value of multiple regression analysis, we omitted this information.

Are the surveys available online? Are they all validated for this kind of examination?

The questionnaire was developed for this study. The headache diagnosis(es), based on the ICHD-3(ß) criteria, were made by a headache expert as part of routine clinical practice before inclusion in the study. Thus, the questionnaire was not used to make a diagnosis but only to collect the clinical data necessary for the study.

We can provide our questionnaire on request (if needed, translated into english).

Reviewer 2 Report

This study evaluates serum levels of selected mitochondria-related biomarkers in patients suffering from episodic and chronic migraine. The authors report several important findings for clinical practice, with the study appropriately design in terms of methods, results presentation and interpretation. However, several issues have to be additionaly addressed:

Introduction

  • Provide more information about the possible mechanism(s) of mitochondrial dysfunction in migraine and its links to specific pathophysiology of migraine; also address whether mitochondrial dysfunction comes first or migraine perhaps induces Mt dysfunction.

Methods

  • How sample size was calculated?
  • How you established that participants had no proved or supposed mitochondrial disorder?
  • Why only FGF21 and GDF15 are selected to be evaluated? Other possible indicators of Mt function in the blood are also available; explain this in more details
  • Provide more information about control group in terms of possible mitochondria-related conditions\

Results

  • Modify Tables 1-3, very hard to follow
  • Conduct an additional analysis comparing experimental and control group through age stages

Discussion

  • Other factors behind migraine (e.g. chronic inflammation) can affect Mt biomarkers and should be accounted for, address this in limitations

Author Response

Provide more information about the possible mechanism(s) of mitochondrial dysfunction in migraine and its links to specific pathophysiology of migraine; also address whether mitochondrial dysfunction comes first or migraine perhaps induces Mt dysfunction.

A: We updated the introduction according to your comment, for which we are very thankful (please see manuscript). We focused on the vasculature and cortical spreading depression as possible links between a mitochondrial dysfunction and migraine. Other concepts are extremely vague and currently do not add a substantial benefit for the reader in our opinion.

How sample size was calculated?

A: As no previous data regarding FGF-21 and GDF-15 levels in migraine are available, no power analysis was possible. Since we expected a much smaller effect/difference than in the mitochondrial disease studies, the patient number should be much higher than in these studies. This goal was achieved with > 200 migraine patients. Recruitment of a "headache-free" control group was significantly more difficult (a common problem in headache trials), so we could not perform 1-1 matching.

If you would like a post-hoc power analysis, we would be happy to perform it. However, the informative value of this analysis is limited.

How you established that participants had no proved or supposed mitochondrial disorder?

A: In addition to the patients' medical history, data were available, for example, in the hospital information system or from physicians' letters brought in by the patients. In addition, a complete clinical-neurological examination was performed. Certainly, mitochondrial diseases cannot be excluded. If there were clinical-syndromal indications of a possible mitochondrial disease, these patients were not included. However, there are also monosymptomatic mitochondrial diseases that we did not cover with this approach. Genetic testing in all participants (e.g., whole-exome sequencing) was not possible given the large number of participants.

With reference to your comment, we have added a point to the limitations as well as to the description of the recruitment in the methods section.

Why only FGF21 and GDF15 are selected to be evaluated? Other possible indicators of Mt function in the blood are also available; explain this in more details

A: In our view, both biomarkers are more sensitive and specific for MD in comparison to other frequently used markers like resting lactate, pyruvate or lactate-pyruvate-ratio (see introduction, line 92-93.). Other parameters (e.g. fatty acid profile or amino acid analysis) were not suitable for this study as they do not serve as screening methods for MD. In addition, FGF-21 and GDF-15 are easy to determine in routine clinical practice, and time-consuming sample preparations are not necessary.

We added a point to the limitation section with respect to the significance of the two markers in MD in general or only in the case of involvement of certain organ systems.

Provide more information about control group in terms of possible mitochondria-related conditions\

A: We refer here to a response to a previous point (checking a proved or supposed mitochondrial disorder).

Results

Modify Tables 1-3, very hard to follow

A: We have chosen the layout for better clarity and have avoided displaying medians. The group-wise analyses were performed in the previous sections. All parameters have then been combined in a multiple regression analysis.

Section 3.4. has been revised, likewise the legends of Fig. 2a and b have been supplemented with reading examples, among other things. We hope that this has improved the readability of the tables.

Conduct an additional analysis comparing experimental and control group through age stages

A: Thank you for this suggestion. We conducted the appropriate nonparametric tests for each group (for GDF-15/age and additionally for FGF-21/BMI) and included a short summary of the results in the specific sections in the results part.

Discussion

Other factors behind migraine (e.g. chronic inflammation) can affect Mt biomarkers and should be accounted for, address this in limitations

A: We added a comment about inflammatory diseases possibly influencing the biomarkers in the limitation section.

Round 2

Reviewer 1 Report

The authors have responded positively to my comments and for that the manuscript is much improved. 

Author Response

Thank you very much for your reply. We will check the language of the manuscript again.

Reviewer 2 Report

The authors improved the paper in line with this reviewer's comments.

Author Response

(The authors gave the same response as above.)
